# Application of a transparent artificial intelligence algorithm for US adults in the obese category of weight

Alexander A. Huang[1]*, Samuel Y. Huang [ORCID][2]*

1 Northwestern University Feinberg School of Medicine, Chicago, Illinois, United States of America,
2 Virginia Commonwealth University School of Medicine, Richmond, Virginia, United States of America

* Alexander.huang@northwestern.edu (AAH); huangs8@vcu.edu (SYH)

## Abstract

### Objective and aims

Identification of associations between the obese category of weight in the general US population will continue to advance our understanding of the condition and allow clinicians, providers, communities, families, and individuals make more informed decisions. This study aims to improve the prediction of the obese category of weight and investigate its relationships with factors, ultimately contributing to healthier lifestyle choices and timely management of obesity.

### Methods

Questionnaires that included demographic, dietary, exercise and health information from the US National Health and Nutrition Examination Survey (NHANES 2017–2020) were utilized with BMI 30 or higher defined as obesity. A machine learning model, XGBoost predicted the obese category of weight and Shapely Additive Explanations (SHAP) visualized the various covariates and their feature importance. Model statistics including Area under the receiver operator curve (AUROC), sensitivity, specificity, positive predictive value, negative predictive value and feature properties such as gain, cover, and frequency were measured. SHAP explanations were created for transparent and interpretable analysis.

### Results

There were 6,146 adults (age > 18) that were included in the study with average age 58.39 (SD = 12.94) and 3122 (51%) females. The machine learning model had an Area under the receiver operator curve of 0.8295. The top four covariates include waist circumference (gain = 0.185), GGT (gain = 0.101), platelet count (gain = 0.059), AST (gain = 0.057), weight (gain = 0.049), HDL cholesterol (gain = 0.032), and ferritin (gain = 0.034).

### Conclusion

In conclusion, the utilization of machine learning models proves to be highly effective in accurately predicting the obese category of weight. By considering various factors such as demographic information, laboratory results, physical examination findings, and lifestyle

**Data Availability Statement:** Data Share Statement: Data described in the manuscript are free available without restriction at: https://wwwn.cdc.gov/nchs/nhanes/continuousnhanes/default.aspx?cycle=2017-2020.

**Funding:** The author(s) received no specific funding for this work.

**Competing interests:** The authors have declared that no competing interests exist.

**Abbreviations:** NCHS, National Center for Health Statistics; NHANES, National Health and Nutrition Examination Surveys; SHAP, Shapely Additive Explanations.

factors, these models successfully identify crucial risk factors associated with the obese category of weight.

## Introduction

Obesity is a worldwide concern, including in the United States, where it has reached epidemic proportions. Over the past few decades, general epidemiological trends indicate a steady increase in the prevalence of obesity across all age groups and socioeconomic strata. This surge in obesity rates has far-reaching consequences for public health, as it is associated with a myriad of serious health issues and imposes a significant economic burden on individuals and society [1].

The alarming rise of obesity in the American population has raised numerous red flags for policymakers, healthcare providers, and researchers. Obesity is not just a matter of aesthetics or body image; it is a multifaceted problem with severe health implications. Individuals affected by obesity are at a higher risk of developing chronic conditions such as type 2 diabetes, cardiovascular diseases, hypertension, and certain cancers [1–4]. Moreover, obesity has been linked to reduced quality of life, decreased life expectancy, and increased healthcare costs.

The biochemical pathways that underlie obesity are complex and multifactorial. Genetic predisposition, environmental factors, sedentary lifestyles, and poor dietary choices all play pivotal roles in the development and progression of obesity. Understanding these underlying mechanisms is crucial for designing effective interventions and tailored treatment strategies.

In response to the obesity epidemic, health authorities have established guidelines and recommendations for its prevention and management. These guidelines typically emphasize a comprehensive approach that includes dietary modifications, increased physical activity, behavior change strategies, and, in some cases, medical interventions. Nevertheless, combating obesity remains a challenge due to its multifaceted nature and the need for personalized interventions [2].

Shapely Additive Explanations (SHAP) has emerged as a promising tool for understanding the complex interplay of factors contributing to obesity. SHAP explanations provide transparent and interpretable insights into the machine learning models used to predict obesity and help identify the most influential features driving the predictions. This enhances our understanding of the complex relationships between various factors and the obese category of weight [5, 6].

The aim of this study is to leverage the power of machine learning, specifically XGBoost, along with SHAP explanations to improve the prediction of the obese category of weight. By utilizing data from the US National Health and Nutrition Examination Survey (NHANES), we seek to investigate the associations between obesity and various demographic, dietary, exercise, and health factors. The ultimate goal is to gain deeper insights into obesity's underlying mechanisms, contribute to the development of more effective preventive strategies, and facilitate timely and personalized management of obesity-related health conditions. By addressing obesity at its root causes, we hope to pave the way for a healthier and more resilient population.

## Methods

A cross-sectional cohort study was conducted with participants who responded to a detailed questionnaire covering demographic information, dietary habits, exercise routines, mental health, as well as laboratory tests and physical examinations using data from the publicly

available National Health and Nutrition Examination Survey (NHANES). The National Center for Health Statistics' (NCHS) Ethics Review Board gave its approval for the study's data gathering and processing. All data, including medical records, survey responses, and demographic data, were de-identified before analysis to safeguard the participants' confidentiality and privacy. Participants gave written agreement prior to the study's start allowing the public release of their data.

## Dataset and cohort selection

The National Center for Health Statistics (NCHS) developed the National Health and Nutrition Examination Survey (NHANES) 2017–2020 to evaluate the health and nutritional status of the American population. The Centers for Disease Control and Prevention (CDC) conducted a comprehensive series of cross-sectional, multi-stage surveys to collect data on health, nutrition, and physical activity for the NHANES dataset. Our investigation focused on adult participants (aged 18 and above) in the NHANES dataset who completed demographic, dietary, exercise, and mental health questionnaires, as well as underwent physical and laboratory examinations. This sample was selected to represent the national population of the United States.

## Assessment of obesity

The obese category of weight is defined as having a Body Mass Index (BMI) greater than or equal to 30. BMI is calculated by dividing an individual's weight (in kilograms) by the square of their height (in meters). A BMI of 30 or above indicates obesity, as per the standard classification set by the World Health Organization (WHO) and many health authorities.

## Model construction and statistical analysis

In this study, the NHANES dataset encompassed covariates that involved socioeconomics, dietary data, actual assessments, research center outcomes, and clinical surveys. Univariate analysis was initially used to explore the relationship between these covariates and the obese category of weight, the outcome variable. To identify strong independent covariates, the machine learning model selected variables with p-values below 0.0001 from the univariate analysis before considering their interaction in the larger model. XGBoost, a widely used and effective algorithm in healthcare predictions, was chosen for this review. Past studies using NHANES data have identified XGBoost as the optimal algorithm, offering a balance of training efficiency, model accuracy, and interpretability. The dataset was split into a train set (80%) and a test set (20%) to determine the parameters for the final model fit. Model performance was evaluated using various parameters, including the area under the receiver operator characteristic curve (AUROC), sensitivity, specificity, positive predictive value, negative predictive value, prevalence, detection rate, detection prevalence, and balanced accuracy. These parameters were used to assess and evaluate the model's performance. Additionally, before carrying out the XGBoost modeling that was present in this paper, other machine learning methods just as artificial neural networks, gradient boost modeling, and random forest, to name a few, were utilized. XGBoost was the most accurate by all metrics (AUROC, sensitivity, specificity) combined and thus was utilized in this paper.

## Model feature importance statistics and SHAP visualization

The Gain metric assesses a feature's importance in the model by calculating its individual contribution for each tree. A higher Gain value suggests greater significance in generating

predictions compared to other features. On the other hand, the Cover metric indicates the relative number of observations associated with a specific feature. It is calculated by summing up the occurrences of that feature across all trees. For example, if feature one appears in 15, 10, 8, and 5 observations in tree one, tree two, tree three, and tree four, respectively, the Cover metric for feature one would be 38 observations. The Cover metric is then expressed as a percentage based on the total cover for all features. Additionally, the Frequency metric represents the relative occurrence of a particular feature in the model's trees. Using the previous example, if feature one appears in 3, 2, 4, and 1 splits within tree one, tree two, tree three, and tree four, respectively, the weightage for feature one would be 10. The Frequency for feature one is determined by calculating its percentage weight relative to the weights of all features.

## Results

Table 1 shows the 6,146 patients that met the inclusion criteria in this study. The average age was 58.39 (SD = 12.94). Individuals had mean HS C-Reactive Protein levels of 4.34 mg/L

Table 1. Demographic variables.

| Total | All Individuals | Obese | Not Obese | |
|---|---|---|---|---|
| Total | 6146 | 2625 | 3521 | |
| Age (years) | 58.39 (12.94) | 57.42 (12.43) | 59.12 (13.27) | p<0.0001 |
| Gender | | | | |
| Female | 3122 (0.51) | 1464 (0.55) | 1678 (0.48) | p<0.0001 |
| Male | 3024 (0.49) | 1181 (0.45) | 1827 (0.52) | p<0.0001 |
| Race/ethnicity | | | | |
| Non-Hispanic White | 2252 (0.37) | 945 (0.36) | 1307 (0.37) | 0.6281 |
| Non-Hispanic Black | 1636 (0.27) | 866 (0.33) | 770 (0.22) | p<0.0001 |
| Hispanic | 1257 (0.2) | 578 (0.22) | 679 (0.19) | 0.0741 |
| Other | 1001 (0.16) | 236 (0.08) | 765 (0.22) | p<0.0001 |
| Creatinine, urine (mg/dL) | 121.99 (80.62) | 134.34 (84.15) | 112.60 (76.52) | p<0.0001 |
| Creatinine, urine (umol/L) | 10784.10 (7126.76) | 11875.98 (7438.47) | 9953.46 (6764.21) | p<0.0001 |
| Urinary Arsenous acid (ug/L) | 0.29 (0.41) | 0.24 (0.36) | 0.32 (0.44) | p<0.0001 |
| Urinary Monomethylarsonic acid (ug/L) | 0.50 (0.60) | 0.42 (0.51) | 0.56 (0.65) | p<0.0001 |
| Direct HDL-Cholesterol (mmol/L) | 1.40 (0.42) | 1.29 (0.35) | 1.48 (0.45) | p<0.0001 |
| Triglyceride (mg/dL) | 114.94 (97.10) | 126.47 (110.39) | 106.50 (85.12) | p<0.0001 |
| Triglyceride (mmol/L) | 1.30 (1.10) | 1.43 (1.25) | 1.20 (0.96) | p<0.0001 |
| Total Cholesterol (mg/dL) | 189.33 (41.85) | 186.59 (41.32) | 191.45 (42.14) | p<0.0001 |
| Total Cholesterol (mmol/L) | 4.90 (1.08) | 4.83 (1.07) | 4.95 (1.09) | p<0.0001 |
| White blood cell count (1000 cells/uL) | 7.20 (5.65) | 7.64 (8.06) | 6.86 (2.51) | p<0.0001 |
| Monocyte percent (%) | 8.29 (2.30) | 8.15 (2.12) | 8.41 (2.43) | p<0.0001 |
| Monocyte number (1000 cells/uL) | 0.58 (0.22) | 0.60 (0.20) | 0.56 (0.23) | p<0.0001 |
| Segmented neutrophils num (1000 cell/uL) | 4.16 (1.70) | 4.40 (1.78) | 3.98 (1.62) | p<0.0001 |
| Eosinophils number (1000 cells/uL) | 0.20 (0.16) | 0.21 (0.16) | 0.19 (0.16) | p<0.0001 |
| Mean cell volume (fL) | 88.97 (6.20) | 87.94 (6.13) | 89.75 (6.14) | p<0.0001 |
| Red cell distribution width (%) | 13.99 (1.38) | 14.22 (1.43) | 13.81 (1.31) | p<0.0001 |
| Platelet count (1000 cells/uL) | 241.76 (65.70) | 247.55 (67.91) | 237.32 (63.60) | p<0.0001 |
| Mean platelet volume (fL) | 8.26 (0.91) | 8.34 (0.90) | 8.20 (0.91) | p<0.0001 |
| Cotinine, Serum (ng/mL) | 56.21 (133.00) | 48.46 (126.35) | 62.19 (137.63) | p<0.0001 |
| Hydroxycotinine, Serum (ng/mL) | 23.75 (66.81) | 19.24 (62.54) | 27.22 (69.74) | p<0.0001 |
| Serum total folate (ng/mL) | 18.38 (12.69) | 17.44 (13.71) | 19.12 (11.77) | p<0.0001 |

(Continued)

**Table 1.** (*Continued*)

| Total | All Individuals | Obese | Not Obese | |
|---|---|---|---|---|
| 5-Methyl-tetrahydrofolate (nmol/L) | 38.69 (23.88) | 36.28 (23.47) | 40.60 (24.04) | p<0.0001 |
| HS C-Reactive Protein (mg/L) | 4.34 (8.89) | 5.89 (9.56) | 3.14 (8.14) | p<0.0001 |
| Mercury, methyl (ug/L) | 1.23 (2.25) | 0.95 (1.46) | 1.46 (2.69) | p<0.0001 |
| Insulin (μU/mL) | 15.13 (25.09) | 21.85 (33.16) | 10.21 (15.11) | p<0.0001 |
| Insulin (pmol/L) | 90.79 (150.55) | 131.09 (198.94) | 61.26 (90.68) | p<0.0001 |
| Iron frozen, Serum (ug/dL) | 85.50 (34.49) | 79.71 (32.67) | 89.97 (35.20) | p<0.0001 |
| UIBC, Serum (umol/L) | 42.17 (11.19) | 43.58 (10.88) | 41.08 (11.30) | p<0.0001 |
| Transferrin Saturation (%) | 27.27 (11.30) | 25.24 (10.39) | 28.84 (11.72) | p<0.0001 |
| Blood lead (ug/dL) | 1.33 (1.30) | 1.15 (1.29) | 1.47 (1.30) | p<0.0001 |
| Blood lead (umol/L) | 0.06 (0.06) | 0.06 (0.06) | 0.07 (0.06) | p<0.0001 |
| Blood cadmium (ug/L) | 0.50 (0.56) | 0.43 (0.54) | 0.55 (0.56) | p<0.0001 |
| Blood cadmium (nmol/L) | 4.43 (4.94) | 3.84 (4.81) | 4.89 (4.99) | p<0.0001 |
| Blood mercury, total (ug/L) | 1.46 (2.60) | 1.14 (1.65) | 1.71 (3.12) | p<0.0001 |
| Blood mercury, total (nmol/L) | 7.29 (13.00) | 5.69 (8.25) | 8.53 (15.59) | p<0.0001 |
| Fasting Glucose (mg/dL) | 117.65 (40.76) | 124.56 (44.12) | 112.60 (37.35) | p<0.0001 |
| Fasting Glucose (mmol/L) | 6.53 (2.26) | 6.91 (2.45) | 6.25 (2.07) | p<0.0001 |
| Albumin, refrigerated serum (g/dL) | 4.03 (0.33) | 3.94 (0.32) | 4.09 (0.32) | p<0.0001 |
| Albumin, refrigerated serum (g/L) | 40.26 (3.30) | 39.45 (3.20) | 40.88 (3.23) | p<0.0001 |
| Bicarbonate (mmol/L) | 25.59 (2.45) | 25.34 (2.49) | 25.77 (2.40) | p<0.0001 |
| Globulin (g/dL) | 3.10 (0.45) | 3.15 (0.44) | 3.06 (0.46) | p<0.0001 |
| Glucose, refrigerated serum (mmol/L) | 5.89 (2.24) | 6.17 (2.42) | 5.67 (2.07) | p<0.0001 |
| Iron, refrigerated serum (ug/dL) | 86.15 (34.36) | 80.34 (32.56) | 90.64 (35.05) | p<0.0001 |
| Lactate Dehydrogenase (LDH) (IU/L) | 161.56 (35.64) | 163.32 (37.29) | 160.20 (34.24) | p<0.0001 |
| Osmolality (mmol/Kg) | 282.10 (5.75) | 282.55 (5.75) | 281.75 (5.72) | p<0.0001 |
| Total Bilirubin (umol/L) | 7.90 (4.58) | 7.42 (4.24) | 8.27 (4.80) | p<0.0001 |
| Total Calcium (mg/dL) | 9.27 (0.39) | 9.24 (0.40) | 9.29 (0.38) | p<0.0001 |
| Total Calcium (mmol/L) | 2.32 (0.10) | 2.31 (0.10) | 2.32 (0.10) | p<0.0001 |
| Cholesterol, refrigerated serum (mg/dL) | 189.64 (41.88) | 186.87 (41.36) | 191.79 (42.16) | p<0.0001 |
| Cholesterol, refrigerated serum (mmol/L) | 4.90 (1.08) | 4.83 (1.07) | 4.96 (1.09) | p<0.0001 |
| Total Protein (g/dL) | 7.13 (0.46) | 7.10 (0.44) | 7.15 (0.47) | p<0.0001 |
| Triglycerides, refrig serum (mg/dL) | 144.36 (107.87) | 155.77 (110.40) | 135.52 (105.05) | p<0.0001 |
| Triglycerides, refrig serum (mmol/L) | 1.63 (1.22) | 1.76 (1.25) | 1.53 (1.19) | p<0.0001 |
| Uric acid (mg/dL) | 5.48 (1.48) | 5.80 (1.51) | 5.24 (1.42) | p<0.0001 |
| Uric acid (umol/L) | 326.02 (88.26) | 344.75 (89.53) | 311.51 (84.47) | p<0.0001 |
| N-ace-S-(3,4-dihidxybutl)-L-cys(ng/mL) | 424.56 (302.67) | 460.12 (317.06) | 398.31 (288.94) | p<0.0001 |
| LBX2DF - Blood 2,5-Dimethylfuran (ng/mL) | 0.03 (0.07) | 0.02 (0.05) | 0.03 (0.08) | p<0.0001 |
| LBXVBZ - Blood Benzene (ng/mL) | 0.07 (0.17) | 0.05 (0.11) | 0.08 (0.21) | p<0.0001 |
| LBXVFN - Blood Furan (ng/mL) | 0.03 (0.03) | 0.03 (0.03) | 0.03 (0.04) | p<0.0001 |
| LBXVIBN - Blood Isobutyronitrile (ng/mL) | 0.04 (0.05) | 0.03 (0.02) | 0.04 (0.06) | p<0.0001 |
| Income_Poverty_Ratio | 2.70 (1.63) | 2.62 (1.62) | 2.76 (1.64) | p<0.0001 |

Descriptive statistics for demographic characteristics and all covariates within the machine learning model, stratified by being in the obese category of weight.

(SD = 8.89), insulin levels of 15.13 umol/mL (SD = 25.09), Blood lead levels of 1.33 ug/dL (SD = 1.30), Blood cadmium levels of 0.50ug/L (SD = 0.56), Uric acid levels of 5.48 mg/dL (SD = 1.48), Creatinine levels of 121.99 mg/dL (SD = 80.62). Compared to those in the obese category of weight to those that were not, there was a mean HS C-Reactive Protein levels of

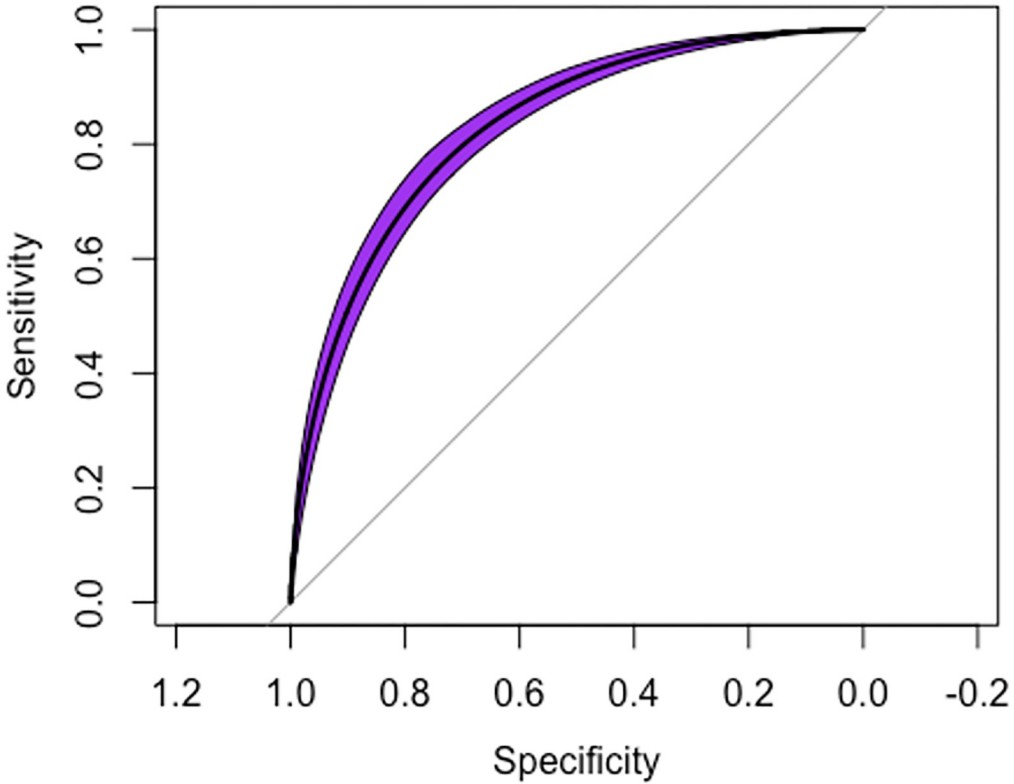

**Fig 1. Receiver operator characteristic curve and model statistics.** The Receiver operating characteristic curve for the machine-learning model predicting whether the patient were in the obese category of weight. AUROC = 0.8295 (P<0.0001).

5.89 mg/L (SD = 9.56) compared to a mean HS C-Reactive Protein levels of 3.14 mg/L (SD = 8.14), insulin levels of 21.85 umol/mL (SD = 33.16) compared to insulin levels of 10.21 umol/mL (SD = 15.11), Blood lead levels of 1.15 ug/dL (SD = 1.29) compared to Blood lead levels of 1.47 ug/dL (SD = 1.30), Blood cadmium levels of 0.43ug/L (SD = 0.54) compared to Blood cadmium levels of 0.55ug/L (SD = 0.56), Uric acid levels of 5.80 mg/dL (SD = 1.51) compared to Uric acid levels of 5.24 mg/dL (SD = 1.42), Creatinine levels of 134.34 mg/dL (SD = 84.15) compared to Creatinine levels of 112.60 mg/dL (SD = 76.52).

**Table 2.**

| Metric | Value |
| --- | --- |
| Sensitivity | 0.9125 |
| Specificity | 0.5615 |
| Positive Predictive Value (Precision) | 0.4226 |
| Negative Predictive Value | 0.9481 |
| Prevalence | 0.2602 |
| Detection Rate (True Positive Rate) | 0.2374 |
| Detection Prevalence | 0.5618 |
| Balanced Accuracy | 0.737 |

Table 2 displays the statistical model metrics evaluated by sensitivity, specificity, positive predictive value (PPV), negative predictive value (NPV), Prevalence, Detection Rate, Detection Prevalence, and Balanced Accuracy.

**Table 3. Model gain statistics.**

| Feature | Gain | Cover | Frequency |
|---|---|---|---|
| HS C-Reactive Protein (mg/L) | 0.181 | 0.103 | 0.049 |
| Insulin (μU/mL) | 0.149 | 0.118 | 0.059 |
| Blood lead (ug/dL) | 0.056 | 0.047 | 0.046 |
| Blood cadmium (ug/L) | 0.038 | 0.039 | 0.037 |
| Uric acid (mg/dL) | 0.037 | 0.054 | 0.042 |
| Creatinine, urine (mg/dL) | 0.036 | 0.041 | 0.049 |
| MCQ366c - Doctor told you to reduce salt in diet | 0.035 | 0.041 | 0.015 |
| Direct HDL-Cholesterol (mmol/L) | 0.032 | 0.035 | 0.032 |
| Albumin, refrigerated serum (g/dL) | 0.030 | 0.042 | 0.028 |
| Red cell distribution width (%) | 0.029 | 0.039 | 0.032 |
| Age | 0.028 | 0.039 | 0.033 |
| Total Protein (g/dL) | 0.024 | 0.035 | 0.031 |
| Mean platelet volume (fL) | 0.017 | 0.022 | 0.025 |
| Mean cell volume (fL) | 0.017 | 0.014 | 0.026 |
| Glucose, refrigerated serum (mmol/L) | 0.016 | 0.021 | 0.023 |
| White blood cell count (1000 cells/uL) | 0.014 | 0.015 | 0.023 |
| Lactate Dehydrogenase (LDH) (IU/L) | 0.014 | 0.015 | 0.027 |
| Income_Poverty_Ratio | 0.014 | 0.010 | 0.028 |
| Urinary Monomethylarsonic acid (ug/L) | 0.014 | 0.026 | 0.018 |
| MCQ160a - Doctor ever said you had arthritis | 0.014 | 0.026 | 0.011 |
| UIBC, Serum (umol/L) | 0.009 | 0.007 | 0.017 |
| Platelet count (1000 cells/uL) | 0.009 | 0.009 | 0.019 |
| Serum total folate (ng/mL) | 0.009 | 0.006 | 0.018 |
| Blood mercury, total (ug/L) | 0.008 | 0.012 | 0.014 |
| Mercury, methyl (ug/L) | 0.008 | 0.008 | 0.013 |
| Gender | 0.008 | 0.017 | 0.009 |
| Monocyte percent (%) | 0.008 | 0.006 | 0.017 |
| Cotinine, Serum (ng/mL) | 0.007 | 0.005 | 0.012 |
| Triglycerides, refrig serum (mg/dL) | 0.007 | 0.005 | 0.014 |
| Total Cholesterol (mg/dL) | 0.007 | 0.006 | 0.015 |
| MCQ560 - Ever had gallbladder surgery? | 0.007 | 0.010 | 0.006 |
| Cholesterol, refrigerated serum (mg/dL) | 0.006 | 0.006 | 0.011 |
| Fasting Glucose (mg/dL) | 0.006 | 0.004 | 0.012 |
| Segmented neutrophils num (1000 cell/uL) | 0.006 | 0.004 | 0.013 |
| N-ace-S-(3,4-dihidxybutl)-L-cys(ng/mL) | 0.006 | 0.006 | 0.011 |
| Iron, refrigerated serum (ug/dL) | 0.005 | 0.006 | 0.010 |
| Bicarbonate (mmol/L) | 0.005 | 0.006 | 0.012 |
| Osmolality (mmol/Kg) | 0.005 | 0.004 | 0.011 |
| Triglyceride (mg/dL) | 0.005 | 0.004 | 0.010 |
| Transferrin Saturation (%) | 0.005 | 0.005 | 0.009 |
| Iron frozen, Serum (ug/dL) | 0.005 | 0.004 | 0.010 |
| Hydroxycotinine, Serum (ng/mL) | 0.005 | 0.006 | 0.007 |
| 5-Methyl-tetrahydrofolate (nmol/L) | 0.005 | 0.005 | 0.008 |
| MCQ371b - Are you now increasing exercise | 0.004 | 0.011 | 0.006 |
| Monocyte number (1000 cells/uL) | 0.004 | 0.006 | 0.008 |
| Total Calcium (mg/dL) | 0.004 | 0.003 | 0.010 |
| MCQ550 - Has DR ever said you have gallstones | 0.004 | 0.006 | 0.005 |

*(Continued)*

**Table 3.** (Continued)

| Feature | Gain | Cover | Frequency |
|---|---|---|---|
| Globulin (g/dL) | 0.004 | 0.002 | 0.008 |
| MCQ300c - Close relative had diabetes? | 0.004 | 0.003 | 0.005 |
| Eosinophils number (1000 cells/uL) | 0.004 | 0.004 | 0.007 |
| Total Bilirubin (umol/L) | 0.004 | 0.002 | 0.007 |
| MCQ300a - Close relative had heart attack? | 0.003 | 0.005 | 0.005 |
| MCQ300b - Close relative had asthma? | 0.003 | 0.005 | 0.005 |
| MCQ371c - Are you now reducing salt in diet | 0.003 | 0.002 | 0.003 |
| LBXVBZ - Blood Benzene (ng/mL) | 0.003 | 0.004 | 0.005 |
| MCQ010 - Ever been told you have asthma | 0.002 | 0.004 | 0.003 |
| Urinary Arsenous acid (ug/L) | 0.002 | 0.002 | 0.004 |
| Basophils number (1000 cells/uL) | 0.001 | 0.001 | 0.002 |
| LBX2DF - Blood 2,5-Dimethylfuran (ng/mL) | 0.001 | 0.001 | 0.002 |
| SMDANY - Used any tobacco product last 5 days? | 0.001 | 0.005 | 0.001 |
| LBXVFN - Blood Furan (ng/mL) | 0.001 | 0.001 | 0.001 |
| MCQ160b - Ever told had congestive heart failure | 0.001 | 0.001 | 0.002 |
| SMQ681 - Smoked tobacco last 5 days? | 0.001 | 0.001 | 0.001 |
| LBXVIBN - Blood Isobutyronitrile (ng/mL) | 0.000 | 0.001 | 0.001 |
| SMQ690A - Used last 5 days - Cigarettes | 0.000 | 0.000 | 0.000 |

The Gain, Cover, and Frequency of all covariates within the XGBoost model. The Gain represents the relative contribution of the feature to the model and is the most important metric of model importance within this study. Covariates ordered according to the Gain statistic.

The machine learning model had 78 features that were found to be significant on univariate analysis (P<0.0001 used). These were fitted into the XGBoost model, Fig 1 and Table 2 shows an AUROC = 0.8295, Sensitivity = 0.9125, Specificity = 0.5615, Positive predictive value 0.4226, negative predictive value of 0.9481 were observed.

Table 3 shows that the top four covariates ranked by the gain, a measure of the percentage contribution of the covariate to the overall model prediction, were HS C-Reactive Protein (mg/L) (Gain = 0.181), Insulin (uU/mL) (Gain = 0.149), Blood lead (ug/dL) (Gain = 0.056), blood cadmium (ug/L) (Gain = 0.038).

In Fig 2, overall SHAP explanations can be seen for all the statistically significant covariates on univariable regression.

In Fig 3, SHAP visualizations were conducted for the top four continuous covariates by overall SHAP explanations. Trends included a positive association of HS C-reactive Protein and insulin and obesity as well as a negative association between blood lead level and blood cadmium level and obesity.

## Discussion

In this cross-sectional cohort study of US adults, an artificial intelligence algorithm trained on information from the National Health and Nutrition Examination Survey (NHANES) demographic, laboratory, physical examination, and lifestyle factors demonstrated a high predictive accuracy with an area under the receiver operating curve (AUROC) of 0.8295. This indicates that the model was able to strongly predict obesity above what is to be expected of standard chance. The top four covariates that had significant associations with the obese category of weight in the artificial intelligence model based off SHAP value included HS C-Reactive protein, insulin, has doctor told you to reduce salt in the diet and blood lead levels. The top four

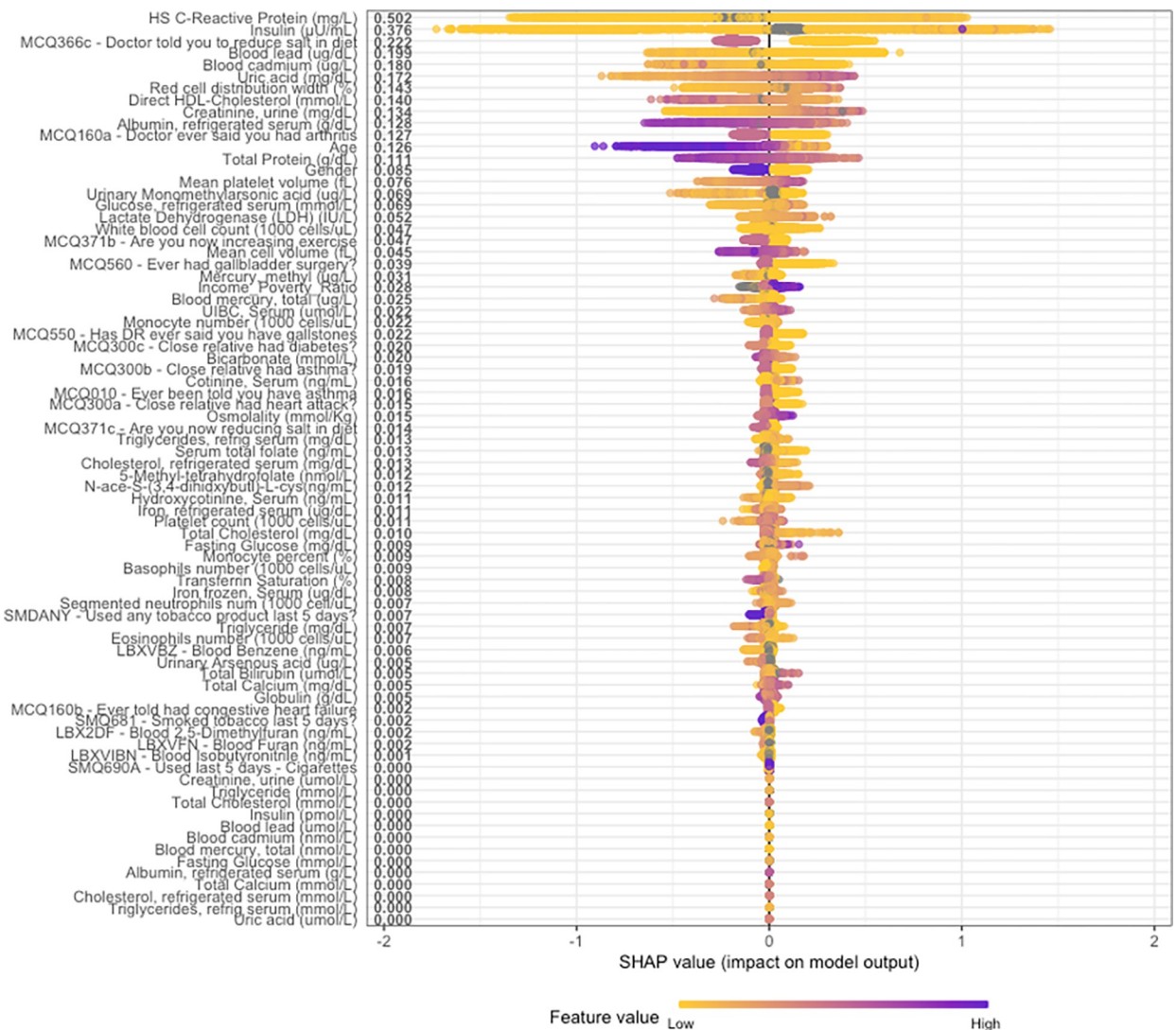

**Fig 2. Overall SHAP explanations.** SHAP explanations, purple color representing higher values of the covariate while yellow representing lower values of the covariate. X-axis is the change in log-odds for advanced the obese category of weight.

covariates ranked by gain, which corresponds to the contribution of each feature in the overall artificial intelligence algorithm includes HS C-Reactive Protein (mg/L) (Gain = 0.181), Insulin (uU/mL) (Gain = 0.149), Blood lead (ug/dL) (Gain = 0.056), blood cadmium (ug/L) (Gain = 0.038).

The artificial intelligence algorithm employed in our study demonstrates consistent associations and directionality with those reported in existing literature concerning the obese category of weight [1, 7–9]. These findings, supported by multiple studies, offer valuable insights into how the algorithm perceives these associations. The alignment of our study's results with established literature enhances our confidence in the algorithm's ability to accurately capture genuine physiological relationships related to obesity. A notable advantage of the algorithmic approach used in our study is its impartiality in identifying significant covariates. By systematically exploring numerous variables based on mathematical relationships, subjective influence from researchers is minimized. This enables the uncovering of nonlinear patterns, and the covariates can be

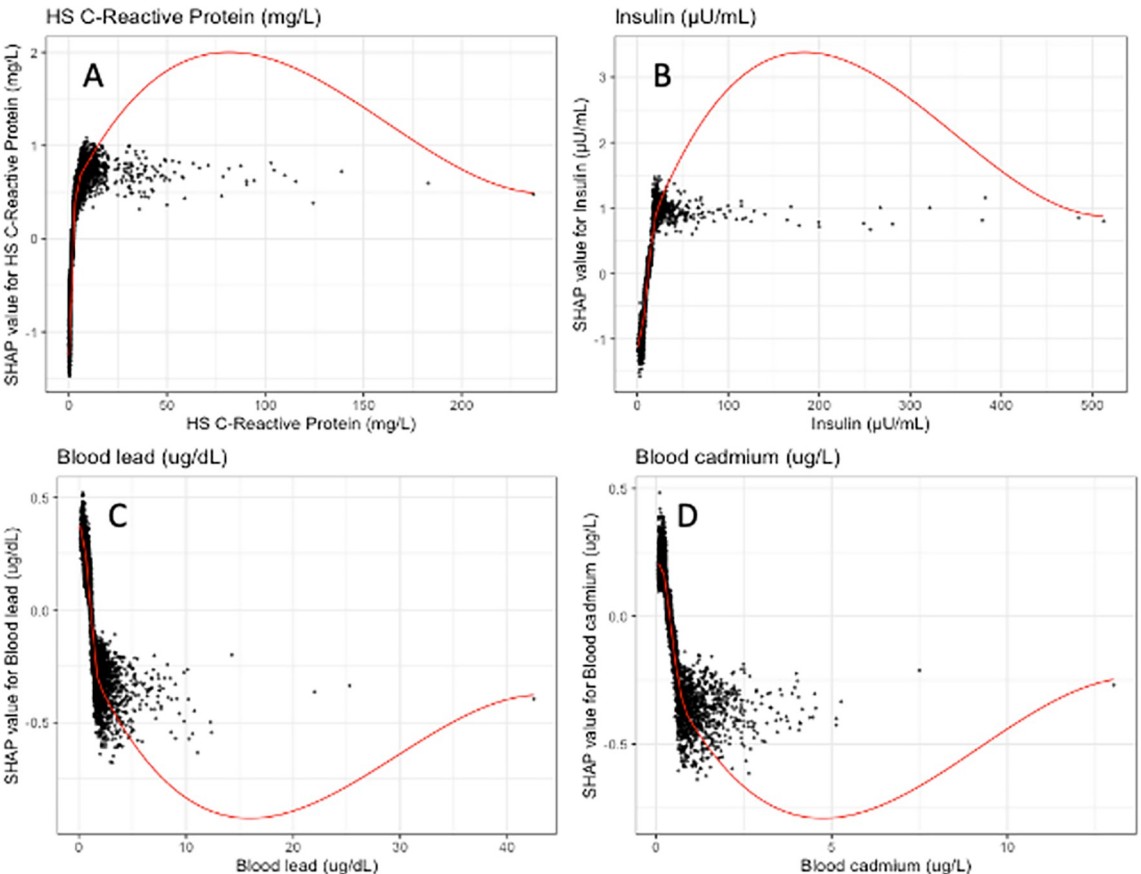

**Fig 3. SHAP explanations, covariate value on the x-axis, change in log-odds on the y-axis, red line represents the relationship between the covariate and log-odds for being in the obese category of weight, each black dot represents an observation.** Covariates: top left–HS C-Reactive Protein (mg/mL), top right–Insulin (uU/mL), bottom left–Blood lead (ug/dL), bottom right–Blood cadmium (ug/L).

ranked based on performance metrics, assessing the overall accuracy and reliability of the machine learning model in predicting obesity [10]. SHAP visualizations further aid researchers in comparing their own understanding of variable relationships with the machine learning model's assessment, allowing for the testing of physiological plausibility. These visualizations provide a valuable tool for validating and comprehending the associations identified by the algorithm, enhancing the interpretability and applicability of the model's predictions [8].

The study inherits the advantages and limitations associated with cross-sectional multistage survey questionnaire studies. These surveys use multistage sampling techniques, ensuring a representative sample and enabling generalizability to the broader population. However, they offer only a snapshot at a single point in time, restricting the ability to establish causality or temporal sequences. Despite their cost-effectiveness and efficiency in gathering data from a large number of participants within a relatively short period, there is a risk of recall and response bias. Therefore, it is vital to take these limitations into account when interpreting findings from cross-sectional multistage survey questionnaire data. Additionally, multiple variables that are significant may not seem to have causal links, such as lead levels and obesity, and further research is needed to identify if these are just correlations in which the cause is a third-variable (for lead levels and obesity–socioeconomic status), or if there may be causal nature. Further studies are needed to confirm these connections.

## Conclusion

The artificial intelligence algorithm predicted obesity over and above random chance and uncovered associations and relationships in a understandable way for clinicians.

## Author Contributions

**Conceptualization:** Samuel Y. Huang.

**Data curation:** Alexander A. Huang.

**Formal analysis:** Alexander A. Huang.

**Methodology:** Alexander A. Huang.

**Project administration:** Alexander A. Huang, Samuel Y. Huang.

**Software:** Alexander A. Huang.

**Supervision:** Alexander A. Huang.

**Writing – original draft:** Samuel Y. Huang.

**Writing – review & editing:** Samuel Y. Huang.

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
