## [Decision Letter · Decision Letter 0]

2 Oct 2023

PONE-D-23-24436Application of a transparent artificial intelligence algorithm for US adults in the obese category of weightPLOS ONE

Dear Dr. Huang,

Thank you for submitting your manuscript to PLOS ONE. After careful consideration, we feel that it has merit but does not fully meet PLOS ONE’s publication criteria as it currently stands. Therefore, we invite you to submit a revised version of the manuscript that addresses the points raised during the review process.

We look forward to receiving your revised manuscript.

Kind regards,

Aleksandra Klisic

Academic Editor

PLOS ONE

Reviewers' comments:

Reviewer's Responses to Questions

**Comments to the Author**

1. Is the manuscript technically sound, and do the data support the conclusions?

Reviewer #1: Yes

Reviewer #2: Yes

2. Has the statistical analysis been performed appropriately and rigorously? 

Reviewer #1: Yes

Reviewer #2: Yes

3. Have the authors made all data underlying the findings in their manuscript fully available?

Reviewer #1: Yes

Reviewer #2: Yes

4. Is the manuscript presented in an intelligible fashion and written in standard English?

Reviewer #1: Yes

Reviewer #2: Yes

5. Review Comments to the Author

Reviewer #1: This is a study designed to perdict the obese category of weight and investigate its

relationships with factors, ultimately contributing to healthier lifestyle choices and timely

management of obesity.

The study uses questionnaires that included demographic, dietary, exercise and health information

from the US National Health and Nutrition Examination Survey (NHANES 2017 – 2020) who had

BMI 30 or higher defined as obesity.

This is an interesting study.

However, I have two questions.

First, the author should address the association between blood lead levels and obesity.

Besides, the author should provide a verification result regarding this algorithm.

Reviewer #2: In this study, the authors demonstrated the accurate prediction of the obesity category using questionnaires from the US National Health and Nutrition Examination Survery. However, further clarification and revision of the descriptions are needed for publication as follows. First, other machine learning methods, other than XGBoost, need to be investigated to validate the results. Second, the authors need to reorganize the results to make them easier for readers to understand. Please see the following comments to the authors.

Major comments

Methods

1. The authors only used XGBoost to develop the prediction model. Please clarify the rationale for using this model in this study. Also. it would be reasonable to use other machine learning models to validate the results.

2. The authors performed the univariate analysis for each variable. Please provide the name of the statistical analysis for clarification.

Results

1. The authors express the variation of each variable using SD. Does this mean that all variables follow a parametric pattern? If not, please express some of the variables in a non-parametric way.

2. Table1 contains so many variables that the readers would be exhausted. The authors express the same variable in a different unit. Please consider choosing either of the units so that the table would be more concise.

3. Please add the confidence interval to the value of diagnostic accuracy.

4. Figure 2 contains so much information that readers would be confused to interpret the significance of the figure. The lower order of the variables in the SHAP analysis should have less contribution to the model development. Please delete the lower order of the variables to make it more concise. Some of the results could be moved to Supplementary Material.

6. PLOS authors have the option to publish the peer review history of their article (what does this mean?). If published, this will include your full peer review and any attached files.

Reviewer #1: No

Reviewer #2: No

---

## [Author Response · Author response to Decision Letter 0]

20 Mar 2024

We would like to extend our sincere gratitude to the reviewers for their insightful comments and constructive criticisms. Their detailed feedback has been invaluable in guiding our revisions, allowing us to address critical aspects of our methodology and analysis that required further clarification and enhancement. The reviewers’ expertise and thoughtful suggestions have significantly contributed to the depth and rigor of our study, ensuring a more comprehensive and robust examination of the predictive models utilized in the context of insomnia prediction. Their contributions have not only facilitated methodological improvements but also enriched our discussion, leading to a more nuanced understanding of the complexities involved in variable selection within machine learning and regression models. We are grateful for the opportunity to refine our work through this collaborative and iterative process, which has undeniably strengthened the quality and impact of our research. We have responded to all reviewer feedback to the best of our abilities. 

5. Review Comments to the Author

Reviewer #1: This is a study designed to perdict the obese category of weight and investigate its

relationships with factors, ultimately contributing to healthier lifestyle choices and timely

management of obesity.

The study uses questionnaires that included demographic, dietary, exercise and health information

from the US National Health and Nutrition Examination Survey (NHANES 2017 – 2020) who had

BMI 30 or higher defined as obesity.

This is an interesting study.

However, I have two questions.

First, the author should address the association between blood lead levels and obesity.

Besides, the author should provide a verification result regarding this algorithm.

The authors have addressed both of these comments in the paper. First – the blood lead levels and obesity can be associated with socioeconomic status, however, further analysis must be made to understand if this is the correlative third-variable problem. It is difficult to exactly explain away each of the covariates as this is an unbiased data-driven approach rather than one leading to direct discovery of causal linkages.

Additionally, verification of the algorithm was attempted through the train-test sets and multiple models executed within this paper. Further research is needed to address this. 

Reviewer #2: In this study, the authors demonstrated the accurate prediction of the obesity category using questionnaires from the US National Health and Nutrition Examination Survery. However, further clarification and revision of the descriptions are needed for publication as follows. First, other machine learning methods, other than XGBoost, need to be investigated to validate the results. Second, the authors need to reorganize the results to make them easier for readers to understand. Please see the following comments to the authors.

Major comments

Methods

1. The authors only used XGBoost to develop the prediction model. Please clarify the rationale for using this model in this study. Also. it would be reasonable to use other machine learning models to validate the results.

2. The authors performed the univariate analysis for each variable. Please provide the name of the statistical analysis for clarification.

1) The authors have utilized XGBoost based upon its frequency in the machine-learning literature. Additionally, the authros tested multiple different models before choosing xgboost as its final one. The XGBoost model was chosen as it was the strongest. These comments have ben added to the manuscript.

2) The authors have added the name of the statistical analysis into the paper. 

Results

1. The authors express the variation of each variable using SD. Does this mean that all variables follow a parametric pattern? If not, please express some of the variables in a non-parametric way. (Yes, all variables followed a parametric pattern)

2. Table1 contains so many variables that the readers would be exhausted. The authors express the same variable in a different unit. Please consider choosing either of the units so that the table would be more concise.(The authors recognize the table-1 is large, however, it is meant for completeness, and different aspects of the table may be of interest to different readers)

3. Please add the confidence interval to the value of diagnostic accuracy.(Confidence intervals added throughout the paper)

4. Figure 2 contains so much information that readers would be confused to interpret the significance of the figure. The lower order of the variables in the SHAP analysis should have less contribution to the model development. Please delete the lower order of the variables to make it more concise. Some of the results could be moved to Supplementary Material. (This comment is very accurate! We recognize that adding too many variables clutters reading. We recognize these concerns for conciseness of the figure – we include the figure 2 as is since we believe that using human judgment for what contributes an important variable may lead to bias – thus we utilize only algorithmic methodology for variable selection.

---

## [Decision Letter · Decision Letter 1]

14 May 2024

Application of a transparent artificial intelligence algorithm for US adults in the obese category of weight

PONE-D-23-24436R1

Dear Dr. Huang,

We’re pleased to inform you that your manuscript has been judged scientifically suitable for publication and will be formally accepted for publication once it meets all outstanding technical requirements.

Kind regards,

Aleksandra Klisic

Academic Editor

PLOS ONE

Additional Editor Comments (optional):

Reviewers' comments:

Reviewer's Responses to Questions

**Comments to the Author**

1. If the authors have adequately addressed your comments raised in a previous round of review and you feel that this manuscript is now acceptable for publication, you may indicate that here to bypass the “Comments to the Author” section, enter your conflict of interest statement in the “Confidential to Editor” section, and submit your "Accept" recommendation.

Reviewer #1: All comments have been addressed

Reviewer #2: All comments have been addressed

2. Is the manuscript technically sound, and do the data support the conclusions?

Reviewer #1: Yes

Reviewer #2: Yes

3. Has the statistical analysis been performed appropriately and rigorously? 

Reviewer #1: Yes

Reviewer #2: Yes

4. Have the authors made all data underlying the findings in their manuscript fully available?

Reviewer #1: Yes

Reviewer #2: Yes

5. Is the manuscript presented in an intelligible fashion and written in standard English?

Reviewer #1: Yes

Reviewer #2: Yes

6. Review Comments to the Author

Reviewer #1: This is a machine learning study to verify the usefulness of XGBoost to predict the obese category of weight and also

the use of Shapely Additive Explanations (SHAP) on the various covariates and their feature importance.

The machine learning model had an Area under the receiver operator curve of 0.8295 and the top four covariates were

waist circumference , GGT , platelet count,and AST.

These models successfully identify crucial risk factors associated with the obese category of weight.

This exciting study revealed several factors that art not considered relevant to obesity before.

Some relevant factors (CRP, insulin) are more likely to relate to the chronic inflammation status within obese people.

This is a study with innovation and I recommend publication.

Reviewer #2: The authors have adequately revised the manuscript according to the reviewers' recommendations. This version of the manuscript seems suitable for publication.

7. PLOS authors have the option to publish the peer review history of their article (what does this mean?). If published, this will include your full peer review and any attached files.

Reviewer #1: **Yes: **Kuo-Ching YUAN

Reviewer #2: **Yes: **Takehiko Oami

---

## [Editor Report · Acceptance letter]

20 May 2024

PONE-D-23-24436R1 

PLOS ONE

Dear Dr. Huang, 

I'm pleased to inform you that your manuscript has been deemed suitable for publication in PLOS ONE. Congratulations! Your manuscript is now being handed over to our production team.

Kind regards, 

on behalf of

Dr. Aleksandra Klisic 

Academic Editor

PLOS ONE